# The nonlinear association between triglyceride glucose-body mass index and femoral neck BMD in nondiabetic elderly men: NHANES 2005-March 2020

Xiuping Xuan[1☯], Rong Sun[2☯], Caibi Peng[3], Lijuan Liu[4], Tiantian Huang[4], Chenghu Huang[4]*

1 Department of Endocrinology, The First Affiliated Hospital of Guangxi Medical University, Nanning, 530021, Guangxi, China, 2 Department of Ophthalmology, Taihe Hospital, Hubei University of Medical, Shiyan, 442000, Hubei, China, 3 Bishan Maternity and Child Hospital of Chongqing, Bishan, Chongqing, 402760, China, 4 Department of Endocrinology, Bishan Hospital of Chongqing, Bishan Hospital of Chongqing Medical University, Bishan, Chongqing, 402760, China

☯ These authors contributed equally to this work.
* tigerhchh@126.com

**Data Availability Statement:** All relevant data are within the manuscript and its Supporting Information files.

## Abstract

### Background

The triglyceride glucose-body mass index (TyG-BMI) has been considered a surrogate marker for assessing insulin resistance. We aimed to correlate the TyG-BMI, triglyceride glucose combined with body mass index, with femoral neck bone mineral density (FN BMD) in non-diabetic elderly men.

### Methods

Using data from the National Health and Nutrition Examination Survey (NHANES) database, totally, 1182 eligible men aged ≥ 50 years without diabetes were included in the current study. Smoothed curves were obtained by a two-piecewise linear regression model and the threshold effects were explored by using a smoothing function.

### Results

TyG-BMI was positive related with and FN BMD with or without adjustment for confounders. However, no typical dose-dependent positive association between TyG-BMI and FN BMD was observed across the TyG-BMI tertiles, indicating a non-linear association. Further analysis by the weighted two-piecewise linear regression model and recursive algorithm suggested that per SD increase in TyG-BMI increased FN BMD by 0.266 gm/cm² when TyG-BMI lower than 168.20. However, when TyG-BMI is higher than 168.20, FN BMD only increased 0.046 gm/cm² for per SD increase of TyG-BMI after fully adjustment (OR = 11.258, 95%CI: 6.034, 16.481). Moreover, subgroup analyses showed that higher TyG-BMI levels were related to elevated FN BMD in all groups, suggesting the consistency of the positive association within these stratas.

**Funding:** Chenghu Huang received funding from the Chongqing Natural Science Foundation (No. CSTB2022NSCQ-MSX1096) and the Scientific and Technological Project of Chongqing Bishan (No. BSKJ2021002). The funders had no role in study design, data collection and analysis, decision to publish, or preparation of the manuscript.

**Competing interests:** The authors declared that there are no conflicts of interest exist.

## Conclusions

This study demonstrated that TyG-BMI is positively associated with FN BMD in a nonlinear fashion among elderly men without diabetes, which may be a reliable marker for the early identification of individuals with lower FN BMD.

## Introduction

Osteoporosis is a systemic skeleton disease characterized by reduced bone mineral density (BMD) and the microarchitectural disruption of bone tissue that leads to increased bone fragility and increased fracture risk [1]. Femoral neck osteoporotic fractures are in high risk of developing severe symptoms and poor prognosis, with the increase of the aging population, and the incidence of femoral neck osteoporotic fractures has risen rapidly [2,3]. And femoral neck bone mineral density (FN BMD) is negatively associated with mortality in the general population and considered useful for predicting proximal femoral fracture [2]. In 2015, the prevalence of osteoporosis at the age of 50 years or more, was 6.8% in men and 22.5% in women [4]. In 2002, the National Osteoporosis Foundation (NOF) reported approximately 10 million US adults aged 50 years and older had osteoporosis and an additional 33 million had low bone mass [5]. These estimates were based on FN BMD. Thus, multiple US and international organizations recommend that clinicians consider FN BMD to diagnose osteoporosis and low bone mass [6]. However, clear understanding of the associated characteristics between osteoporosis and body metabolism has not been fully revealed.

It is clear that insulin signaling plays a critical role in the regulation of bone acquisition [7,8]. Insulin signaling modulates both bone formation by osteoblasts and bone resorption by osteoclasts [9]. The insulin receptor in osteoblasts is required for osteoblast proliferation, survival and differentiation [10]. Whether insulin resistance (IR) can affect bone remain unclear. In experimental studies, hyperinsulinemia and IR may contribute to reduced bone turnover, the inhibition of acquisition of glucose by the skeleton and deficits in bone microarchitecture [10,11,12]. Nearly all clinical data on the association between IR and bone have come from observational studies. It's an inconvenient method to assess insulin sensitivity by the euglycemic-hyperinsulinemic clamp method—the gold standard measure of insulin sensitivity [13]. In practice, the current commonly used assessment of IR is the homeostasis model assessment of insulin resistance (HOMA-IR), a combined measure of fasting plasma glucose (FPG) and insulin [13]. Nevertheless, there has been disagreement on the association between HOMA-IR and BMD [14–18].

Recently, the triglyceride and glucose (TyG) index, derived from the combination of fasting triglyceride (TG) and FPG, has been proposed as an effective substitute for IR [19]. In addition, the TyG-BMI, a modified TyG index combining obesity indices such as body mass index (BMI), has been suggested because obesity is closely linked to IR [19]. Moreover, the TyG-BMI may act as a better marker for assessing IR than HOMA-IR [19,20]. However, investigations into the association between the TyG-BMI and BMD are extremely limited.

Due to the higher incidence rate of osteoporosis in women aged 50 years and over, most studies focus on elderly women and less on men. Thus, the present study evaluated the association between the TyG-BMI and FN BMD after adjustment for possible confounding factors in nondiabetic elderly men in the continuous National Health and Nutrition Examination Survey (NHANES).

## Methods and materials

### Study population and sampling

The source of the data for this study was the continuous NHANES, a survey conducted by the National Center for Health Statistics of the Centers for Disease Control and Prevention (https://wwwn.cdc.gov/nchs/nhanes/). NHANES data were collected from a nationally representative sample of a civilian noninstitutionalized US population using a multistage probability design. Questionnaire, laboratory testing, and physical examination data are available. Human Research Ethics Committee approval was not needed, as all data used in the analysis were deidentified and made publicly available.

The exclusion criteria were as follows: (1) women; (2) participants aged < 50 years; (3) participants with hypoglycemia (FPG ≤ 50 mg/dl or 2.8 mmol/L) or diabetes (FPG ≥ 110 mg/dl or 7.0 mmol/L) at baseline; and (4) participants with diseases including cancer or malignant diseases, diabetes, thyroid diseases, chronic renal failure, inflammatory arthritis, and chronic liver disease and the use of drugs in the treatment of dyslipidemia, corticosteroids, sex hormones and diuretics. Furthermore, participants who had incomplete FN BMD data and invalid data were excluded from the final model.

### Assessments of BMD

In the present study, the FN BMD data were analyzed because the femoral neck has been proposed as the reference skeletal site for defining osteoporosis in epidemiological studies [21]. BMD (gm/cm$^2$) measurements were obtained using dual-energy X-ray absorptiometry (DXA) with Hologic QDR 4500A fan-beam densitometers (Hologic Inc., Bedford, MA, USA) [22,23].

### Definition of TyG-BMI terms

BMI was calculated as weight/height$^2$. After an overnight fast, venous samples were also obtained through aseptic techniques for phosphorus, total calcium, plasma glucose and lipid measurements.

Definitions of the TyG-BMI terms were calculated as follows:

1. TyG index = Ln [TG (mg/dL) × FPG (mg/dL)/2] [12].

2. TyG-BMI = TyG index × BMI (kg/m$^2$)

### Covariates

The association between the TyG-BMI and BMD was calculated using FN BMD data collected in the NHANES from 2005– March 2020, but the 2011–2012 and 2015–2016 surveys were not included due to the unavailability of FN BMD data. We merged the continuous NHANES 2005– March 2020 and to ensure a large and representative sample. Sex was classified as "man" or "woman"; race/ethnicity was categorized as "Mexican American", "non-Hispanic White", "non-Hispanic Black", or "other race/ethnicity"; education level was classified as "less than high school", "high school diploma", or "more than high school"; marital status was categorized as "married or living with partner" or "single". Of health-related variables, smoking was categorized as follows: individuals with a lifetime smoking history of at least five packs of cigarettes (equivalent to 100 cigarettes) or more were classified as 'smokers,' and those with a smoking experience of less than five packs of cigarettes and were currently nonsmoking were designated 'nonsmokers.' Alcohol drinking was binary with individuals consuming alcohol once a month or more for the past year classified as "drinkers", and others as "nondrinkers".

Diabetes mellitus was defined as "doctor told you had diabetes" or "taking diabetic pills to lower blood sugar now" or "taking insulin now". Cancer or malignancy was defined as "ever doctor told you had cancer or malignancy"; thyroid was defined as "do you still have thyroid problem"; chronic liver disease was defined as "told you had any liver condition"; family of osteoporosis was defined as "parents ever told had osteoporosis". Inflammatory arthritis was defined as "rheumatoid arthritis" or "psoriatic arthritis"; diuretics included hydrochlorothiazide bumetanide chlorthalidone triamterene, torsemide, spironolactone, indapamide, furosemide; drugs for hyperlipemia included atorvastatin, fluvastatin, lovastatin, pitavastatin, pravastatin, rosuvastatin, simvastatin, ezetimibe, fenofibrate, fenofibric acid, niacin and omega-3 polyunsaturated fatty acids. Physical activity was categorized by metabolic equivalent hours per week (MET-min/wk) of moderate-to-vigorous physical activity and was defined as "less active ($< 600$ MET min/week)" or "active ($\geq 600$ MET min/week)" [24]. From 2000 to 2006, serum total 25 (OH)D was measured by radioimmunoassay (RIA; DiaSorin) [25,26]. Beginning in 2007–2008, a fully validated standardized liquid chromatography-tandem mass spectrometry (LC-MS/MS) method was used to measure 25 (OH)$D_3$ and 25-hydroxyvitamin $D_2$ [25 (OH)$D_2$] for all eligible participants. And we converted the 2005–2006 vitamin D data to equivalent 25 (OH)D measurements from the standardized (LC-MS/MS) method according to the previous study [26]. Beginning in 2007–2008, a fully validated standardized liquid chromatography-tandem mass spectrometry (LC-MS/MS) method was used to measure 25 (OH)$D_3$ for all eligible participants.

## Statistical analysis

Statistical analysis was performed according to the guidelines of the Centers for Disease Control and Prevention (CDC) (https://wwwn.cdc.gov/nchs/nhanes/tutorials/ default.aspx). We calculated all estimates accounting for NHANES sample weights and excluded the outliers of the TyG-BMI by box plots. For missing values of each categorical covariate, we set the missing values as another level of the categorical covariate to be adjusted in the model. For missing values of each continuous covariate, since it was acceptable to continue our analysis without further evaluation or adjustment if 10% or less of the data were missing, we did not make any changes.

Firstly, to determine the association between TyG-BMI and FN BMD, continuous variables are presented as mean ± standard deviation (SD) or as median (low tertile–intermediate tertile—high tertile). Kolmogorov-Smirnov tests were used to check data normality. For continuous normally distributed variables, statistical differences were evaluated using Student's t test or the Kolmogorov-Smirnov test is used to check the normality of the data. Categorical data are presented as frequencies percentages. Statistical differences between categorical variables were analyzed by Chi-square tests. Weighted linear regression models were used to calculate the differences in continuous variables between participants without events and those with events while weighted $\chi 2$ tests were used to calculate the differences in categorical variables except data release cycle between them.

Secondly, to investigate whether there is a nonlinear association of TyG-BMI with FN BMD, fitted curves of association were plotted using restricted cubic spline functions without adjustment. TyG-BMI was used as the x variable and FN BMD was used as the y variable. If a significant nonlinear association was determined by the logarithm likelihood ratio test, the turning point of TyG-BMI can be calculated using a recursive algorithm and two-piecewise.

Thirdly, to examine whether TyG-BMI are independently associated with FN BMD, odds ratios (ORs) and 95% confidence intervals (CIs) for TyG-BMI and FN BMD were analyzed by multivariate logistic regression models. Crude model was not adjusted; and multivariate-

adjusted Model 1 was adjusted for age. Covariates were included as potential confounders in the final models if they changed the estimates of TyG-BMI by more than 10% or were significantly associated with FN BMD. The following covariates were selected a priori on the basis of established associations and/or plausible biological relations and tested: the following covariates were selected a priori on the basis of established associations and/or plausible biological relations and tested: age, ethnicity, education, family of osteoporosis, smoking and drinking status, systolic blood pressure, the concentration of total cholesterol (TC), low-density lipoprotein (LDL), high-density lipoprotein (HDL) in multivariate-adjusted Model 2. Considering the classical risks on osteoporosis [27,28], we simultaneously adjusted for levels of phosphorus, total calcium, 25 $(OH)D_3$ in multivariate-adjusted Model 2. Furthermore, smooth curve fittings and generalized additive models were used to address the nonlinear relationship between TyG-BMI and BMD. For nonlinear models, the inflection point in the relationship between TyG-BMI and FN BMD was calculated using a recursive algorithm, with a two-piecewise linear regression model conducted on both sides of the inflection point, when nonlinearity was detected.

All statistical analyses were performed with the statistical software R (http://www.R-project.org, The R Foundation) and EmpowerStats software (http://www.empowerstats.com, X&Y Solutions, Inc., Boston, MA). A p value less than 0.05 (two-sided) was considered statistically significant.

## Results

### Study participant characteristics

In the current study, a total of 1182 eligible participants across the NHANES from 2007 to March 2020 were included for the analysis and the exclusion criteria are presented in Fig 1. The clinical and biochemical characteristics of the population studied are presented in Table 1. The levels of diastolic blood pressure (DBP), FPG, TG, low-density lipoprotein cholesterol (LDL), total cholesterol (TC), and FN BMD were the highest in the participants within the high TyG-BMI tertile (all P values <0.05). The percentage of marriage or men living with partner, smoker, men of Mexican American descent and men with a BMI $\geq$ 30 kg/m$^2$ were also the highest across the high TyG-BMI tertile. In contrast, the concentration of high-density lipoprotein cholesterol (HDL) was the lowest in the high TyG-BMI tertile. Additionally, levels of serum calcium, phosphorus, and 25 $(OH)D_3$, the status of education, drinking, and physical activity, the ratio of family history of osteoporosis, age, and the percentage of men aged $\geq$ 65 years did not show any significant differences among TyG-BMI tertiles (Table 1).

### Multiple regression model

The correlation coefficient between the TyG-BMI and FN BMD was 0.321 (P < 0.001). In multiple linear regression analysis, FN BMD increased by 0.043 gm/cm$^2$ (95% CI: 0.036, 0.050) in the crude model, 0.040 gm/cm$^2$ (95% CI: 0.033, 0.048) in multivariate-adjusted Model 1, and 0.058 gm/cm$^2$ (95% CI: 0.045, 0.072) in multivariate-adjusted Model 2 with each SD increase in the TyG-BMI (Table 2).

The TyG-BMI was then categorized into three tertiles. The crude model, multivariate-adjusted Model I, or multivariate-adjusted Model II did not show any typical dose-dependent association between the TyG-BMI tertile and BMD. For the multivariate-adjusted Model II, compared to that of the low TyG-BMI tertile, the OR for FN BMD was 5.671 (95% CI: 2.843, 8.499) in the intermediate TyG-BMI tertile, whereas it was 10.993 (95% CI: 10.993 (7.930, 14.057) in the high TyG-BMI tertile. These findings suggested the existence of a nonlinear association between the TyG-BMI and BMD (Table 2).

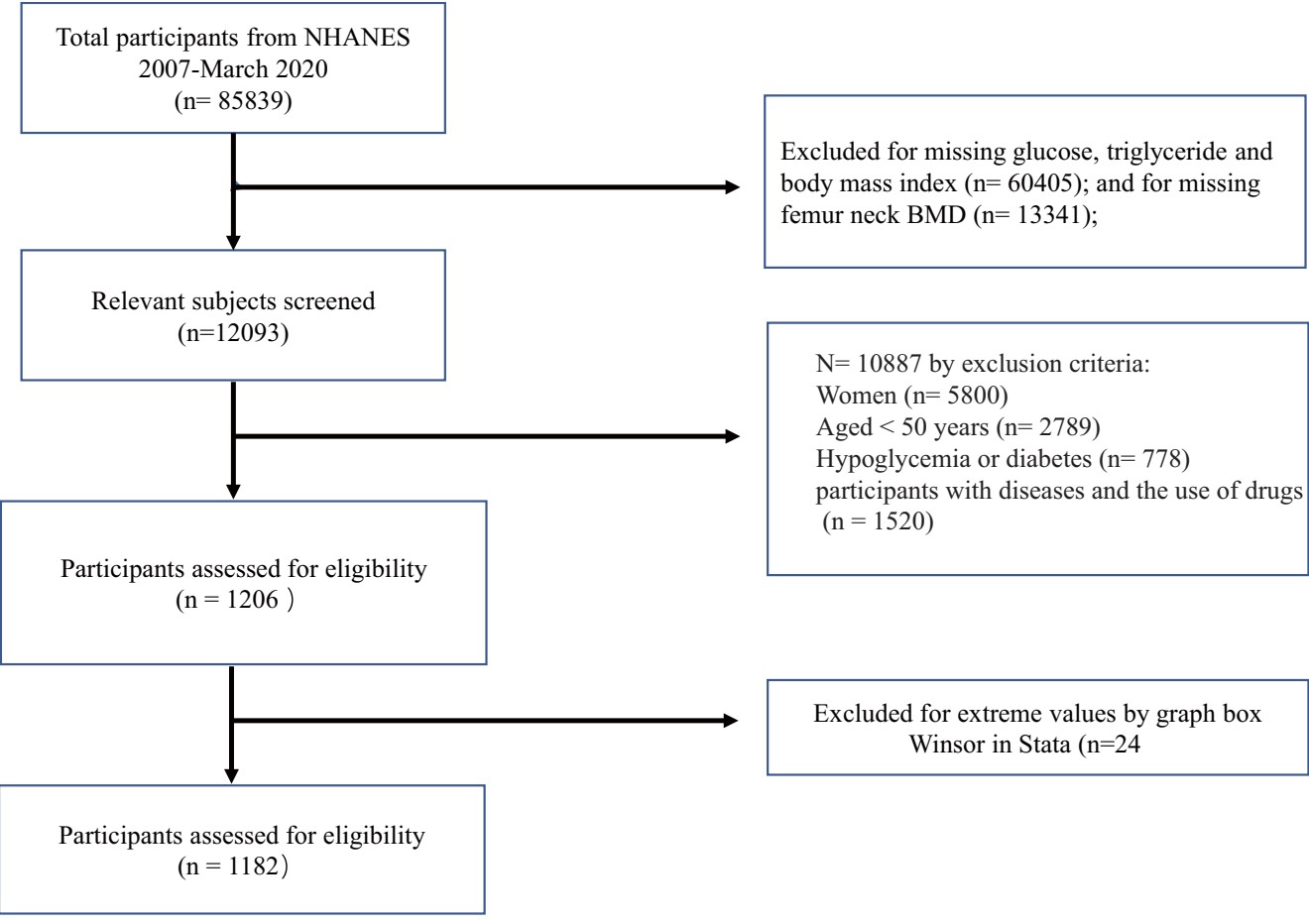

**Fig 1. Flow chart showing the selection of the study sample from the continuous National Health and Nutrition Examination Survey (NHANES).** All analyses presented in this paper were based on 1182 surveyed participants, each with complete anthropometric, blood-based indicator, and covariable data.

## Weighted two-piecewise linear regression model analysis using a smoothing function

We further used smooth curve fitting (penalty curve method) to explore the potential nonlinear relationship of the TyG-BMI and FN BMD. As shown in Fig 2, the association of the TyG-BMI with FN BMD was nonlinear with or without adjustment. Using the weighted two-piecewise linear regression model and recursive algorithm, we found that the inflection point of the TyG-BMI was 168.20 with the likelihood ratio test (P = 0.003). Interestingly, each SD increase in the TyG-BMI led to an increase in the FN BMD by 0.266 gm/cm$^2$ (95% CI: 0.128, 0.404) when the TyG-BMI was lower than 168.20. However, when the TyG-BMI was higher than 168.20, the FN BMD only increased by 0.046 gm/cm$^2$ (95% CI: 0.036, 0.055) gm/cm$^2$ per SD increase in the TyG-BMI after full adjustment (OR = 11.258, 95% CI: 6.034, 16.481) (Table 3).

## Subgroup analysis

In the subgroup analysis, there were positive associations between the TyG-BMI and FN BMD in the crude model and multivariate-adjusted models in almost all subgroups of age, race/ethnicity, education level, marital status, smoking and drinking status, family history of

**Table 1. Baseline characteristics of the participants.**

| Variables | TyG-BMI | | | P value * |
|---|---|---|---|---|
| | **T1 (low)** | **T2 (intermediate)** | **T3 (high)** | |
| N, % | 394 | 395 | 393 | |
| TyG-BMI | 185.56 ± 18.98 | 230.37 ± 11.12 | 283.40 ± 27.60 | <0.001 |
| Age (years) | 60.86 ± 8.87 | 59.85 ± 8.43 | 60.17 ± 7.43 | 0.222 |
| 50–64 | 233 (59.14%) | 263 (66.58%) | 263 (66.92%) | |
| ≥ 65 | 161 (40.86%) | 132 (33.42%) | 130 (33.08%) | |
| Race/Ethnicity, N (%) | | | | < 0.001 |
| Mexican American | 27 (6.85%) | 53 (13.42%) | 75 (19.08%) | |
| Other Hispanic | 31 (7.87%) | 41 (10.38%) | 46 (11.70%) | |
| Non-Hispanic White | 163 (41.37%) | 176 (44.56%) | 167 (42.49%) | |
| Non-Hispanic Black | 118 (29.95%) | 83 (21.01%) | 84 (21.37%) | |
| Other Race | 55 (13.96%) | 42 (10.63%) | 21 (5.34%) | |
| Education, N (%) | | | | 0.295 |
| Less than high school | 116 (29.44%) | 105 (26.58%) | 121 (30.79%) | |
| High school diploma | 86 (21.83%) | 83 (21.01%) | 98 (24.94%) | |
| More than high school | 192 (48.73%) | 206 (52.15%) | 174 (44.27%) | |
| Marital status, N (%) | | | | 0.014 |
| Married or living with partner | 264 (67.01%) | 279 (70.63%) | 300 (76.34%) | |
| Single | 130 (32.99%) | 116 (29.37%) | 93 (23.66%) | |
| Drinking status, N (%) | | | | 0.288 |
| Yes | 211 (53.55%) | 238 (60.25%) | 222 (56.49%) | |
| No | 78 (19.80%) | 76 (19.24%) | 81 (20.61%) | |
| Smoking status, N (%) | | | | <0.001 |
| Non-smokers | 149 (37.82%) | 169 (42.78%) | 169 (43.00%) | |
| Smokers | 133 (33.76%) | 83 (21.01%) | 53 (13.49%) | |
| Physical activity, N (%) | | | | 0.825 |
| Less active | 178 (45.18%) | 170 (43.04%) | 175 (44.53%) | |
| Active | 216 (54.82%) | 225 (56.96%) | 218 (55.47%) | |
| Family of osteoporosis, N (%) | | | | 0.275 |
| Yes | 30 (7.61%) | 29 (7.34%) | 27 (6.87%) | |
| No | 345 (87.56%) | 337 (85.32%) | 350 (89.06%) | |
| PIR | 3.11 ± 1.60 | 3.18 ± 1.54 | 3.61 ± 1.54 | <0.001 |
| SBP (mmHg) | 126.53 ± 17.31 | 126.81 ± 16.55 | 127.72 ± 15.23 | 0.580 |
| DBP (mmHg) | 71.56 ± 13.85 | 73.77 ± 10.10 | 74.41 ± 11.58 | 0.003 |
| BMI (kg/m$^2$) | 22.73 ± 2.15 | 26.99 ± 1.67 | 32.44 ± 3.14 | <0.001 |
| < 18.5 | 25 (6.35%) | 0 (0.00%) | 0 (0.00%) | |
| ≥18.5, <25 | 318(80.71%) | 52(13.16%) | 0 (0.00%) | |
| ≥25, <30 | 51 (12.94%) | 322 (81.52%) | 112 (28.50%) | |
| ≥30 | 0 (0.00%) | 21 (5.32%) | 281 (71.50%) | |
| FPG (mmol/L) | 5.50 ± 0.47 | 5.74 ± 0.46 | 5.87 ± 0.45 | <0.001 |
| Triglycerides (mmol/L) | 0.89 ± 0.41 | 1.27 ± 0.62 | 1.52 ± 0.79 | <0.001 |
| LDL (mmol/L) | 2.90 ± 0.74 | 3.31 ± 0.85 | 3.14 ± 0.80 | <0.001 |
| TC (mmol/L) | 4.88 ± 0.83 | 5.21 ± 0.94 | 5.07 ± 0.90 | <0.001 |
| HDL (mmol/L) | 1.56 ± 0.41 | 1.32 ± 0.33 | 1.24 ± 0.30 | <0.001 |
| Serum calcium (mmol/L) | 2.33 ± 0.08 | 2.33 ± 0.08 | 2.32 ± 0.10 | 0.104 |
| Serum phosphorus (mmol/L) | 1.10 ± 0.16 | 1.08 ± 0.16 | 1.07 ± 0.16 | 0.083 |
| Serum 25(OH)D$_3$ (ng/mL) | 70.92 ± 27.18 | 67.03 ± 20.26 | 65.97 ± 19.88 | 0.106 |

*(Continued)*

**Table 1.** (Continued)

| Variables | TyG-BMI | | | P value * |
|---|---|---|---|---|
| | **T1 (low)** | **T2 (intermediate)** | **T3 (high)** | |
| FN BMD (gm/cm$^2$) | 0.76 ± 0.12 | 0.80 ± 0.12 | 0.85 ± 0.12 | <0.001 |

NOTs

Continuous variables are presented as the mean ± standard deviation (normal distribution), and P value was calculated by weighted linear regression model.

Categorical variables are presented d as number (%), and * P value was calculated by weighted chi-square test.

Notes: BMD, bone mineral density; BMI, body mass index; DBP, diastolic blood pressure; FPG, fasting plasma glucose; FN BMD, Femoral Neck BMD; HDL-C, high-density lipoprotein cholesterol; LDL-C, low-density lipoprotein cholesterol; PIR, family income to poverty ratio; SBP, systolic blood pressure; TC, Total cholesterol; TyG-BMI, triglyceride and glucose-body mass index.

osteoporosis, PIR, physical activity level, total cholesterol, HDL, LDL, plasma calcium, plasma phosphorus, 25 (OH)D$_3$, SBP and DBP (Fig 3). Importantly, there was no significant interaction effect between the TyG-BMI in the subgroup analysis in both the crude and multivariate-adjusted models, further verifying the consistency of the positive effects of the TyG-BMI on FN BMD in nondiabetic elderly men.

## Discussion

To the best of our knowledge, our study is the first to discover that the TyG-BMI, a novel marker of IR, was significantly associated with FN BMD in nondiabetic elderly men. The positive association between the TyG-BMI and FN BMD was nonlinear (Fig 2 and Table 3).

Insulin resistance, manifesting as hyperinsulinemia, is the pivotal pathogenic component for the development of metabolic syndrome, including obesity, dyslipidemia, elevated blood pressure and altered glucose metabolism [29]. As the testing method for IR using the euglycemic-hyperinsulinemic clamp method or HOMA-IR is expensive and inconvenient[13], the identification of simple and practical clinical markers for IR has come to the fore. Under this background, a large number of obesity-related indicators and anthropometric parameters have been investigated, such as HOMA-IR, TyG, and TyG-related parameters. Among them,

**Table 2. Univariate and multivariate effects of the association between TyG-BMI on FN BMD (mg/cm$^2$) in non-diabetic elderly men in NHANES.**

| | Crude model | | Multivariate-adjusted Model 1 | | Multivariate-adjusted Model 2 | |
|---|---|---|---|---|---|---|
| | **β (95% CI)** | **P-value** | **β (95% CI)** | **P-value** | **β (95% CI)** | **P-value** |
| TyG-BMI | | | | | | |
| per SD increase (continued) | 0.043 (0.036, 0.050) | < 0.001 | 0.040 (0.033, 0.048) | < 0.0001 | 0.058 (0.045, 0.072) | < 0.001 |
| TyG-BMI tertile | | | | | | |
| T1: < 212.61 (low) | Ref. | | Ref. | | Ref. | |
| T2: 212.61–250.15 (intermediate) | 3.478 (1.670, 5.287) | < 0.001 | 2.918 (1.143, 4.693) | 0.013 | 5.671 (2.843, 8.499) | < 0.001 |
| T3: ≥ 250.16 (high) | 8.707 (6.896, 10.518) | < 0.001 | 8.058 (6.278, 9.837) | < 0.001 | 10.993 (7.930, 14.057) | < 0.001 |

NOTs.

Crude model: None.Multivariate-adjusted Model 1: Age.

Multivariate-adjusted Model 2: Age; race/ethnicity; education; marital status; drinking status; smoking status; SBP (Smooth); DBP (Smooth); TC; HDL-C; LDL-C; family of osteoporosis; physical activity; PIR; serum calcium; serum phosphorus; serum 25(OH)D$_3$.

BMD, bone mineral density; BMI, body mass index; DBP, diastolic blood pressure; FN BMD, Femoral Neck BMD; HDL-C, high-density lipoprotein cholesterol; LDL-C, low-density lipoprotein cholesterol; PIR, family income to poverty ratio; SBP, systolic blood pressure; TC, Total cholesterol; TyG-BMI, triglyceride and glucose- body mass index.

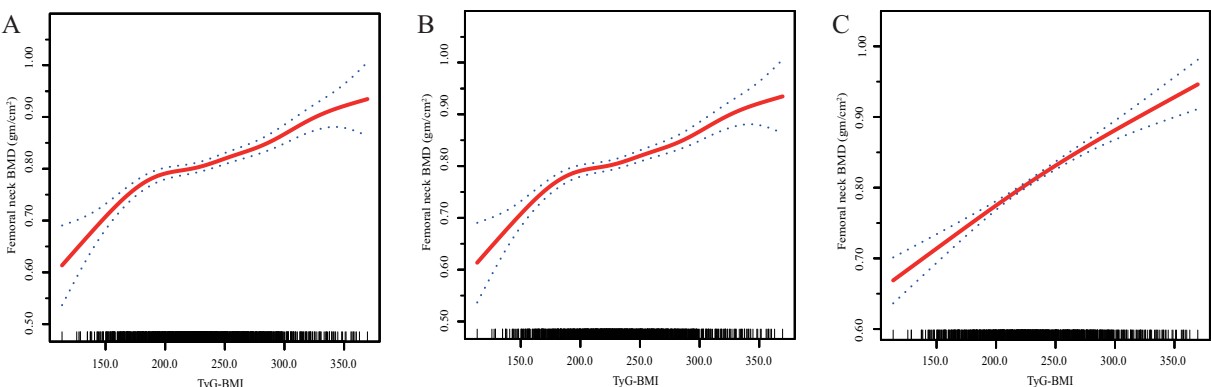

**Fig 2.** Smooth curve fitting of associations between TyG-BMI and femoral neck BMD in NHANES in non-diabetic elderly men in crude model (A), multivariate-adjusted Model 1 (B) and multivariate-adjusted Model 2 (C). Solid red line represents the smooth curve fit between variables. Blue bands represent the 95% of confidence interval from the fit.

the TyG-BMI, one of the TyG-related parameters, is a combined variable composed of FPG, TG and BMI levels, which was first reported by Professor Er in 2016 [12].

Nearly all clinical data on the association between IR and BMD are derived from observational studies. Although controversial, the association between IR and BMD has been investigated. Choi YJ. et al. reported that no significant association between IR, assessed by HOMA-IR, and trabecular or cortical volumetric BMD was observed in adolescent populations in an analysis of the Korea and US National Health and Nutrition Examination Surveys (NHANES) [30]. However, in participants with IR-related conditions, such as polycystic ovary syndrome and lipodystrophy, BMD was reduced [31,32]. Similarly, studies also showed that IR, evaluated using either HOMA-IR or TyG index, was inversely associated with BMD [16,33]. In contrast, Cherif R et al. found that BMD was improved in obese and insulin-

**Table 3. Threshold effect analysis of TyG-BMI on FN BMD (gm/cm$^2$) using weighted piece-wise linear regression in elderly non-diabetic men in NHANES.**

|  | Crude model | Multivariate-adjusted Model 1 | Multivariate-adjusted Model 2 |
|---|---|---|---|
| TyG-BMI |  |  |  |
| Inflection point (K = 168.2) |  |  |  |
| per SD increase |  |  |  |
| ≤ K (continued) | 0.158(0.060, 0.257) * | 0.162(0.064, 0.259) * | 0.266 (0.128, 0.404) ** |
| > K (continued) | 0.038 (0.030, 0.047) ** | 0.035 (0.027, 0.043) ** | 0.046 (0.036, 0.055) ** |
| OR (95% CI) |  |  |  |
| ≤ K | Ref | Ref | Ref |
| > K | 10.424 (7.567, 13.281) * | 10.085 (7.298, 12.871) * | 11.258 (6.034, 16.481) * |
| P for trend | 0.024 | 0.014 | 0.003 |

NOTs

Crude model: None.

Multivariate-adjusted Model 1: Age.

Multivariate-adjusted Model 2: Race/ethnicity; education; marital status; drinking status; smoking status; SBP (Smooth); DBP (Smooth); TC; HDL-C; LDL-C; family of osteoporosis; physical activity; PIR; serum calcium; serum phosphorus; serum VitD3.

BMD, bone mineral density; BMI, body mass index; DBP, diastolic blood pressure; FN BMD, Femoral Neck BMD; HDL-C, high-density lipoprotein cholesterol; LDL-C, low-density lipoprotein cholesterol; PIR, family income to poverty ratio; SBP, systolic blood pressure; TC, Total cholesterol; TyG-BMI, body mass index; VitD3: 25(OH) D$_3$.

* P < 0.01; ** P < 0.001.

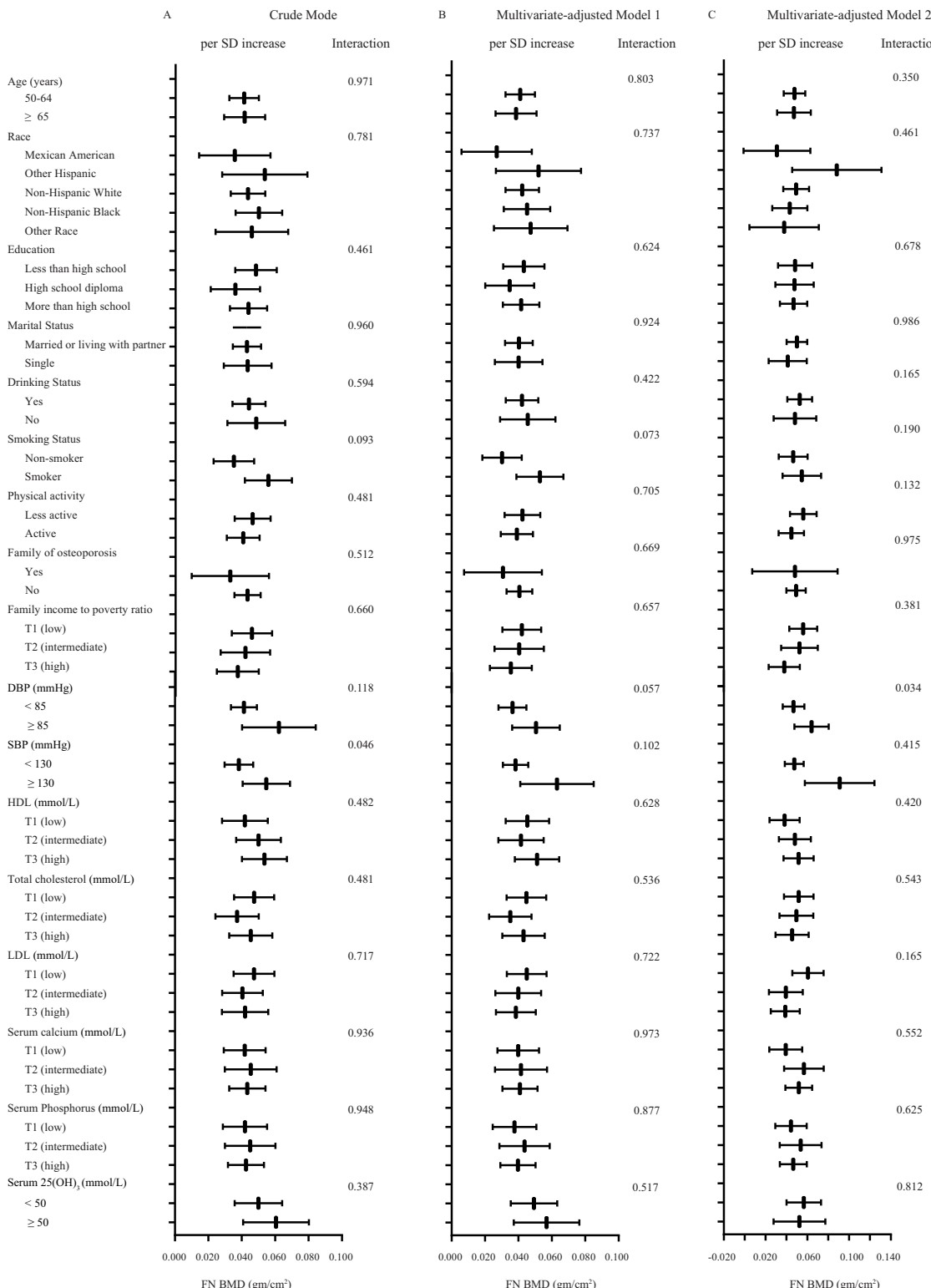

**Fig 3. Subgroup analysis of the associations between TyG-BMI and femoral neck BMD in NHANES in non-diabetic participants by age, race/ethnicity, education, marital status, drinking status, smoking status, SBP, DBP, TC, HDL-c, LDL-c, family of osteoporosis, physical activity, PIR, FPG, triglyceride, BMI, serum calcium, serum phosphorus, serum 25 (OH)D3.** Crude model adjusted for: None. Multivariate-adjusted Model 1 adjusted for: Age. Multivariate-adjusted Model 2 adjusted for: Age, race/ethnicity, education, marital status, smoking and drinking status, family of osteoporosis, PIR, physical activity, systolic blood pressure, HDL cholesterol, LDL cholesterol. The box represents per SD in FN BMD, and the line represents the 95% CI.

resistant Tunisian postmenopausal women [34]. Moreover, in nondiabetic postmenopausal women, IR, assessed by HOMA-IR, is also accompanied by increased areal or volumetric BMD [14,15]. In addition, even in the same study, metabolic syndrome, a grouping of pathological conditions related to IR, resulted in a significantly increased positive association with BMD in men but not in women, revealing a sex difference [18]. The only study on the association reported that TyG index was also inversely associated with FN BMD in non-diabetic men aged ≥ 50 years and postmenopausal women in Korea [33]. Paradoxically, in our study, the association between TyG index and FN BMD missed (Supplementary Table 1). Interestingly, compared with other quantitative metrics to assess IR, the TyG-BMI may be a better marker for assessing IR in individuals with IR-related metabolic diseases [19,20,35–37]. This led us to ask: what is the correlation between the TyG-BMI and BMD?

To solve this question, we explored the association between the TyG-BMI and FN BMD in nondiabetic elderly men using the NHANES 2005-March 2020 data. In our study, there was a threshold effect in the relationship between the TyG-BMI and FN BMD using the weighted two-piecewise linear regression model and recursive algorithm. Each SD increase in the TyG-BMI increased FN BMD by 0.266 gm/cm$^2$ (95% CI: 0.128, 0.404) when the TyG-BMI was lower than 168.20. However, when the TyG-BMI was higher than 168.20, FN BMD increased by only 0.046 gm/cm$^2$ per SD increase in the TyG-BMI after full adjustment (OR = 11.258, 95% CI: 6.034, 16.481) (Fig 2, Table 3). These results first suggested a nonlinear positive correlation between IR and BMD, which is partly consistent with the positive association in previous studies [14,15,18].

We also confirmed that the nonlinear positive correlation between TyG-BMI on FN BMD persisted in nondiabetic elderly men. In the baseline, some variables (such as the ratio of race/ethnicity) varied with the levels of TyG-BMI (Table 1). However, in almost all subgroups, the correlation between TyG-BMI and FN BMD was nonlinear and positive, with no significant interaction effect in the crude model and multivariate-adjusted models (Fig 3).

Although the underlying mechanism of the independent association between the TyG-BMI and FN BMD is unclear, it may be related to obesity, the vital component of the TyG-BMI. Obesity is traditionally linked to higher BMD and reduced fracture risk [38,39]. This could be due to the elevation of serum insulin-like growth factor-1 (IGF-1) levels in obese individuals [40]. Serum IGF-1 directly regulates bone growth and density [41], promoting bone growth and remodeling [42]. Another mechanism to explain the increased BMD in overweight/obese children is the biomechanical load of body weight in weighing-bearing bone, which in turn stimulates bone formation [39,43]. Moreover, increased insulin levels, which are frequently accompanied by IR, may be an important factor for the elevated BMD under an insulin-resistant status. Clinical data suggested that elevated insulin levels are usually associated with an increased BMD [44,45]. For example, Zhou H et al. found that BMD increased by 0.49 mg/cm$^2$ in response to each unit increment in log-transformed fasting insulin [45]. Similarly, Stolk R et al. suggested that the mean age-adjusted BMD (mg/cm$^2$) increased by 0.35 in men and by 0.25 in women with each mU/L postload insulin increase [44]. Additionally, FPG and TGs, the other two components of the TyG-BMI, may also play a role in the correlation between the TyG-BMI and BMD, considering their positive effects on BMD [45–48].

The main strength of our study is that the measurement of IR by the TyG-BMI was convenient and cost-effective and might be widely used clinically. Our study also has limitations. First, although our study used nationally representative US Census population data and combined sample weights for statistical analysis, the number of participants was limited and the sample size was small. Second, because of the observational design, causality cannot be inferred. Third, our study population was elderly males; therefore, the findings may not be generalizable to females or other age groups. Finally, the current study focused on the effects

of the TyG-BMI on BMD at the femoral neck and found a positive association. However, whether this association existed at other skeletal sites remained unclear.

## Conclusion

The TyG-BMI, derived from the TyG index, was positively related to FN BMD in nondiabetic elderly men, with a nonlinear association. The TyG-BMI was the more significant mediator of the associations between IR and FN BMD. These findings improved our understanding of the plausible relationship between IR and BMD and provide some clues for adjusting for the confounding effects of metabolic markers when studying the relationship of IR and osteoporosis.

## Supporting information

**S1 Table. Univariate and multivariate effects of the association between TyG index on FN BMD (mg/cm$^2$) in non-diabetic elderly men in NHANES.**
(DOCX)

**S1 File.**
(XLSX)

## Author Contributions

**Conceptualization:** Xiuping Xuan, Caibi Peng, Chenghu Huang.

**Data curation:** Lijuan Liu, Tiantian Huang, Chenghu Huang.

**Formal analysis:** Rong Sun, Chenghu Huang.

**Funding acquisition:** Chenghu Huang.

**Investigation:** Rong Sun, Lijuan Liu, Tiantian Huang, Chenghu Huang.

**Methodology:** Rong Sun, Lijuan Liu, Tiantian Huang, Chenghu Huang.

**Project administration:** Chenghu Huang.

**Resources:** Chenghu Huang.

**Software:** Chenghu Huang.

**Writing – original draft:** Xiuping Xuan, Lijuan Liu, Tiantian Huang, Chenghu Huang.

**Writing – review & editing:** Xiuping Xuan, Chenghu Huang.

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
