## [Decision Letter · Decision Letter 0]

20 Feb 2023

PONE-D-22-30746The nonlinear association between triglyceride glucose-body mass index and femoral neck BMD in nondiabetic elderly men: NHANES 2005-March 2020PLOS ONE

Dear Dr. Huang,

Thank you for submitting your manuscript to PLOS ONE. After careful consideration, we feel that it has merit but does not fully meet PLOS ONE’s publication criteria as it currently stands. Therefore, we invite you to submit a revised version of the manuscript that addresses the points raised during the review process.

It is an interesting study where authors investigated the association between triglyceride glucose-body mass index and femoral neck BMD in nondiabetic elderly men by using a larger data. However, some technical issues should be addressed further as mentioned by the reviewers. Moreover, the participants were limited due to selection criteria so the results should be interpreted with caution.

We look forward to receiving your revised manuscript.

Kind regards,

Shaonong Dang, PhD

Academic Editor

PLOS ONE

Journal Requirements:

2. "PLOS requires an ORCID iD for the corresponding author in Editorial Manager on papers submitted after December 6th, 2016. Please ensure that you have an ORCID iD and that it is validated in Editorial Manager. To do this, go to ‘Update my Information’ (in the upper left-hand corner of the main menu), and click on the Fetch/Validate link next to the ORCID field. This will take you to the ORCID site and allow you to create a new iD or authenticate a pre-existing iD in Editorial Manager. Please see the following video for instructions on linking an ORCID iD to your Editorial Manager account: " ext-link-type="uri" xlink:type="simple">https://www.youtube.com/watch?v=_xcclfuvtxQ"

3. Thank you for including your ethics statement:  "N/A".  

1. For studies reporting research involving human participants, PLOS ONE requires authors to confirm that this specific study was reviewed and approved by an institutional review board (ethics committee) before the study began. Please provide the specific name of the ethics committee/IRB that approved your study, or explain why you did not seek approval in this case.

Additional Editor Comments (if provided):

It is an interesting study where authors investigated the association between triglyceride glucose-body mass index and femoral neck BMD in nondiabetic elderly men by using a larger data. However, some technical issues should be addressed further as mentioned by the reviewers. Moreover, the participants were limited due to selection criteria so the results should be interpreted with caution.

Reviewers' comments:

Reviewer's Responses to Questions

**Comments to the Author**

1. Is the manuscript technically sound, and do the data support the conclusions?

Reviewer #1: Partly

Reviewer #2: Yes

Reviewer #3: Partly

2. Has the statistical analysis been performed appropriately and rigorously? 

Reviewer #1: Yes

Reviewer #2: Yes

Reviewer #3: Yes

3. Have the authors made all data underlying the findings in their manuscript fully available?

Reviewer #1: Yes

Reviewer #2: Yes

Reviewer #3: Yes

4. Is the manuscript presented in an intelligible fashion and written in standard English?

Reviewer #1: Yes

Reviewer #2: Yes

Reviewer #3: No

5. Review Comments to the Author

Reviewer #1: Dear Editor

I read interestingly the manuscript entitled “The nonlinear association between triglyceride glucose-body mass index and femoral neck BMD in nondiabetic elderly men: NHANES 2005-March 2020”. The study seems conducted carefully.

- Please prepare a list of abbreviations at the beginning of the Meta DATA.

Abstracts

1) Abstract should be informative, background did not explain the question of this research and the answer which authors search for it

2) Please indicate an exact explanation of TyG-BMI

3) Keywords: are these keywords are Mesh terms? Word that serves as a keyword, as to the meaning of that condition must be a Mesh term

Introduction

-“Multiple US and international organizations recommend that clinicians consider BMD at the

femoral neck (FN) to diagnose osteoporosis and low bone mass”

please expand and explain more about using FN as gold standard to diagnose osteoporosis and low bone mass

-Please correct :

Line 68: Whether insulin resistance (IR) can a-ect bone remain unclear. In experimental

Results

It would be great if the authors prepare a correlation diagram for the study

Discussion

- The authors should list and shortly discuss the limitation of their study, for instance their limited number of participants and small sample size

Reviewer #2: The authors aimed to correlate the triglyceride glucose-body mass index (TyG-BMI), a novel marker for assessing IR with femoral neck (FN) BMD in nondiabetic elderly men. they used data from the National Health and Nutrition Examination Survey (NHANES) database. they results indicated TyG-BMI is positively associated with FN BMD in a nonlinear fashion among elderly men without diabetes. The study is well designed, the statistical modeling supports the results.

minor:

1. Please cosider all abbreviation in text and title and abstract. May some abbreviations were used without mentioning the full expression for the first time. Also check them for tables and graphs too.

2. I believe discussion would be more extended according to the results. I feel this section did not bring all findings under debate. For example, the results based of baseline indicated there would be potentially groups under risk of more or less association for TyG-BMI (for instance according to etnicity,...).

Reviewer #3: The data of this study comes from a large sample and representative survey, however, the research object of this paper is a part of it, and there are many exclusion criteria, which may lead to the problem of under-representation of the study population, and the results may be biased. Table 1 should focus on the distribution characteristics of FN BMD by sex and age in the study population, and describe the consistency and difference with the large sample and representative population in the discussion to comprehensively evaluate the representativeness of the study sample and the reliability of the results.

This study analyzes the relationship between TYG-BMI and FN BMD, and assumes that it is a new indicator. Therefore, it is suggested to analyze the relationship between TYG and FN BMD at the same time and compare them.

Why only elderly men are included in the study and women are excluded is not clearly explained in the introduction and background.

The full text analysis strategy needs to be further reorganized, and the relationship between TYG-BMI and FN BMD needs to be analyzed qualitatively and quantitatively.

6. PLOS authors have the option to publish the peer review history of their article (what does this mean?). If published, this will include your full peer review and any attached files.

Reviewer #1: No

Reviewer #2: **Yes: **Mohamad Amin Pourhoseingholi

Reviewer #3: **Yes: **liying xing

---

## [Author Response · Author response to Decision Letter 0]

27 Sep 2023

Here below is our description on revision according to the letter. 

1. Please ensure that your manuscript meets PLOS ONE's style requirements, including those for file naming. The PLOS ONE style templates can be found at https://journals.plos.org/plosone/s/file?id=wjVg/PLOSOne_formatting_sample_main_body.pdf and https://journals.plos.org/plosone/s/file?id=ba62/PLOSOne_formatting_ sample_title_authors_affiliations.pdf

Reply:

Done accordingly.

2. "PLOS requires an ORCID iD for the corresponding author in Editorial Manager on papers submitted after December 6th, 2016. Please ensure that you have an ORCID iD and that it is validated in Editorial Manager. To do this, go to ‘Update my Information’ (in the upper left-hand corner of the main menu), and click on the Fetch/Validate link next to the ORCID field. This will take you to the ORCID site and allow you to create a new iD or authenticate a pre-existing iD in Editorial Manager. Please see the following video for instructions on linking an ORCID iD to your Editorial Manager account: https://www.youtube.com/watch?v=_xcclfuvtxQ"

Reply:

ORCID ID for the corresponding author (Chenghu Huang) is 0000-0001-7088-1753.

3. For studies reporting research involving human participants, PLOS ONE requires authors to confirm that this specific study was reviewed and approved by an institutional review board (ethics committee) before the study began. Please provide the specific name of the ethics committee/IRB that approved your study, or explain why you did not seek approval in this case.

Reply: 

We already mentioned the ethics statement on Page 4, Lines 116-118. “Human Research Ethics Committee approval was not needed, as all data used in the analysis were deidentified and made publicly available.” 

Response to Reviewer 1:

1. Abstracts

1) Abstract should be informative, background did not explain the question of this research and the answer which authors search for it

Reply: We explained the question on Page 2, Lines 31-34.

2) Please indicate an exact explanation of TyG-BMI

Reply: We explained the TyG-BMI on Page 2, Lines 31-34.

3) Keywords: are these keywords are Mesh terms? Word that serves as a keyword, as to the meaning of that condition must be a Mesh term

Reply: We corrected the keywords on Page 2, Lines 54-55.

2. Introduction

-“Multiple US and international organizations recommend that clinicians consider BMD at the femoral neck (FN) to diagnose osteoporosis and low bone mass”

please expand and explain more about using FN as gold standard to diagnose osteoporosis and low bone mass

Reply: We explained the confusions on Page 3, Lines 60-63 and Lines 66-69.

3. -Please correct :

 Line 68: Whether insulin resistance (IR) can a-ect bone remain unclear. In experimental

Reply: We corrected it on Page 3, Lines 77-78.

4. Results

It would be great if the authors prepare a correlation diagram for the study

Reply: Pearson's correlation coefficient between the TyG-BMI and FN BMD was 0.321 (P 0.001). We analyzed this on Page 9, Lines 235-236.

5. Discussion

- The authors should list and shortly discuss the limitation of their study, for instance their limited number of participants and small sample size

Reply: We listed the limitation of our study on Page 13, Lines 346-348.

Response to Reviewer 2:

Reviewer #2: The authors aimed to correlate the triglyceride glucose-body mass index (TyG-BMI), a novel marker for assessing IR with femoral neck (FN) BMD in nondiabetic elderly men. they used data from the National Health and Nutrition Examination Survey (NHANES) database. they results indicated TyG-BMI is positively associated with FN BMD in a nonlinear fashion among elderly men without diabetes. The study is well designed, the statistical modeling supports the results.

minor:

1. Please consider all abbreviation in text and title and abstract. May some abbreviations were used without mentioning the full expression for the first time. Also check them for tables and graphs too.

Reply: 

We correct those errors.

2. I believe discussion would be more extended according to the results. I feel this section did not bring all findings under debate. For example, the results based of baseline indicated there would be potentially groups under risk of more or less association for TyG-BMI (for instance according to etnicity,...).

Reply: 

We extensively discussed the results on Page 12, Lines 320-325. “We also confirmed that the nonlinear positive correlation between TyG-BMI on FN BMD persisted in nondiabetic elderly men. In the baseline, some variables (such as the ratio of race/ethnicity) varied with the levels of TyG-BMI (Table 1). However, in almost all subgroups, the correlation between TyG-BMI and FN BMD was nonlinear and positive, with no significant interaction eﬀect in the crude model and multivariate-adjusted models (Figure 3).

Response to Reviewer 3:

1. This study analyzes the relationship between TYG-BMI and FN BMD, and assumes that it is a new indicator. Therefore, it is suggested to analyze the relationship between TYG and FN BMD at the same time and compare them. 

Reply: We have already focused on the relationship between TyG and FN BMD. However, because the relationship between them is ambiguous, we did not mention it in the manuscript. In this revised manuscript, I have added this on Page 11, Lines 301-304.

2. Why only elderly men are included in the study and women are excluded is not clearly explained in the introduction and background.

Reply: We clearly explained it in the introduction and background Page 4, Lines 95-96.

4. The full text analysis strategy needs to be further reorganized, and the relationship between TYG-BMI and FN BMD needs to be analyzed qualitatively and quantitatively.

Reply: We quantitatively discussed it on Page 11, Lines 312-316 and qualitatively on Page 12, Lines 319-324.

Many grammatical or typographical errors have been revised.

---

## [Decision Letter · Decision Letter 1]

21 Dec 2023

The nonlinear association between triglyceride glucose-body mass index and femoral neck BMD in nondiabetic elderly men: NHANES 2005-March 2020

PONE-D-22-30746R1

Dear Dr. Huang,

We’re pleased to inform you that your manuscript has been judged scientifically suitable for publication and will be formally accepted for publication once it meets all outstanding technical requirements.

Kind regards,

Shaonong Dang, PhD

Academic Editor

PLOS ONE

Additional Editor Comments (optional):

Authors have addressed the comments from the reviewers, and the manuscript has been improved much for publication.

Reviewers' comments:

Reviewer's Responses to Questions

**Comments to the Author**

1. If the authors have adequately addressed your comments raised in a previous round of review and you feel that this manuscript is now acceptable for publication, you may indicate that here to bypass the “Comments to the Author” section, enter your conflict of interest statement in the “Confidential to Editor” section, and submit your "Accept" recommendation.

Reviewer #1: All comments have been addressed

2. Is the manuscript technically sound, and do the data support the conclusions?

Reviewer #1: (No Response)

3. Has the statistical analysis been performed appropriately and rigorously? 

Reviewer #1: (No Response)

4. Have the authors made all data underlying the findings in their manuscript fully available?

Reviewer #1: (No Response)

5. Is the manuscript presented in an intelligible fashion and written in standard English?

Reviewer #1: (No Response)

6. Review Comments to the Author

Reviewer #1: The revision has been made and I am satisfied with the answer provided by the authors and manuscript is suitable now.

7. PLOS authors have the option to publish the peer review history of their article (what does this mean?). If published, this will include your full peer review and any attached files.

Reviewer #1: No

---

## [Editor Report · Acceptance letter]

15 Jan 2024

PONE-D-22-30746R1 

PLOS ONE

Dear Dr. Huang, 

I'm pleased to inform you that your manuscript has been deemed suitable for publication in PLOS ONE. Congratulations! Your manuscript is now being handed over to our production team.

Kind regards, 

on behalf of

Dr. Shaonong Dang 

Academic Editor

PLOS ONE